# Privacy-Shielded Image Compression: Defending Against Exploitation from Vision-Language Pretrained Models

Xuelin Shen [* 2]   Jiayin Xu [* 2 3]   Kangsheng Yin [2 3]   Wenhan Yang [1 †]

## Abstract

The improved semantic understanding of vision-language pretrained (VLP) models has made it increasingly difficult to protect publicly posted images from being exploited by search engines and other similar tools. In this context, this paper seeks to protect users' privacy by implementing defenses at the image compression stage to prevent exploitation. Specifically, we propose a flexible coding method, termed Privacy-Shielded Image Compression (PSIC), that can produce bitstreams with multiple decoding options. By default, the bitstream is decoded to preserve satisfactory perceptual quality while preventing interpretation by VLP models. Our method also retains the original image compression functionality. With a customizable input condition, the proposed scheme can reconstruct the image that preserves its full semantic information. A Conditional Latent Trigger Generation (CLTG) module is proposed to produce bias information based on customizable conditions to guide the decoding process into different reconstructed versions, and an Uncertainty-Aware Encryption-Oriented (UAEO) optimization function is designed to leverage the soft labels inferred from the target VLP model's uncertainty on the training data. This paper further incorporates an adaptive multi-objective optimization strategy to obtain improved encrypting performance and perceptual quality simultaneously within a unified training process. The proposed scheme is plug-and-play and can be seamlessly integrated into most existing Learned Image Compression (LIC) models. Extensive experiments across multiple downstream tasks have demonstrated the effectiveness of our design.

---

[*]Equal contribution [1]Pengcheng Laboratory [2]Guangdong Laboratory of Artificial Intelligence and Digital Economy (SZ) [3]Shenzhen University. Correspondence to: Wenhan Yang <yangwh@pcl.ac.cn>.

*Proceedings of the 42[nd] International Conference on Machine Learning*, Vancouver, Canada. PMLR 267, 2025. Copyright 2025 by the author(s).

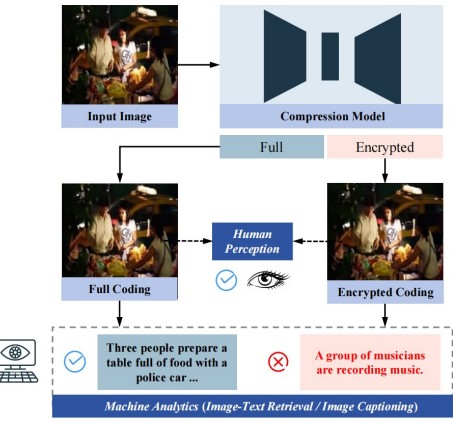

*Figure 1.* In the default protected coding mode, our compression model protects images by preserving content similar to the original, while concealing machine-perceived semantics. Additionally, the model can encode images in a way that maintains both critical pixel fidelity and semantics.

## 1. Introduction

In recent years, the rapid advancement of large-scale Vision-Language Pretrained (VLP) models has transformed traditional task-specific approaches, significantly improving the performance across a range of vision tasks. State-of-the-art VLP models, such as Contrastive Language-Image Pre-training (CLIP) (Radford et al., 2021), A Large-scale ImaGe and Noisy-Text Embedding (ALIGN) (Jia et al., 2021), and InstructBLIP (Dai et al., 2023), demonstrate excellent performance in a general sense for visual information understanding. They perform well in tasks including not only image–text retrieval (Fang et al., 2021; Ma et al., 2022; Luo et al., 2022; Yan et al., 2023), image captioning (Chen et al., 2015), visual question answering (Antol et al., 2015), cross-modality grounding (Peng et al., 2023b), but also conventional vision tasks including image classification (Fu et al., 2022; Peng et al., 2023a; Abdelfattah et al., 2023), facial attribute analysis (Yang et al., 2022; Zhou et al., 2024), and image segmentation (Xu et al., 2022; Chen et al., 2023).

VLP models are highly effective at capturing semantic information, but this strength also comes with risks. Their powerful ability to analyze semantics raises concerns about privacy and data security. Images shared on community platforms can be easily understood by these models, making

it hard to stop them from being indexed by search engines or reused as open data without the user's permission. For instance, in (Johan Modin, 2022), the authors demonstrate the feasibility of tracking the trajectory of a specific vehicle or person from a gallery of surveillance videos by providing only a textual description to the VLP model. Similarly, by incorporating these powerful VLP models, search engines could more easily retrieve relevant images without the need for handcrafted labels. Under these circumstances, there is significant potential to explore privacy protection approaches in the context of defending against VLP models. This emphasizes the need to protect user data and prevent its misuse.

An intuitive idea is to embed certain information into image coding (compression or transcoding) that does not affect visual perception, enabling a universal and efficient way to protect the content of the image. A related research direction focuses on the vulnerability of deep learning models to specific noise patterns. In particular, existing backdoor attack schemes (Yu et al., 2023; 2024) follow a similar approach by embedding invisible triggers into input images. These triggers are designed to degrade compression processes and transform them into adversarial attack patterns, effectively misleading downstream machine vision models. Directly applying these methods to address the proposed new problem faces two key challenges: 1) In the context of designing privacy-protecting compression networks, injecting triggers at the input stage inevitably reduces flexibility and compression efficiency. Specifically, this backdoor attack is an irreversible process, leading to a coding practice that requires separate encoding steps—one with trigger injection and one without—in order to generate distinct bitstreams for attacking and normal purposes. 2) Since an ideal privacy-preserving coding could support both visual perception and machine recognition (may under different coding modes), the challenge lies in efficiently coding both types of information and activating them under specific conditions. This issue has not been explored in previous methods.

In this paper, we focus on Image Compression problem under the consideration of privacy protection problem and propose a novel Privacy-Shielded Image Compression (PSIC) framework. [1] This framework builds on a formulation similar to backdoor attacks and introduces new innovations to significantly improve coding flexibility and efficiency, while also ensuring compactness, human perception compatibility, and encryption requirements, among other valuable attributes. In detail, the proposed PSIC enables a single bitstream to be decoded into two distinct versions through a conditional trigger injection module takes the mode indicator (*encrypted* or *full*) as an additional input: the *encrypted version*, produced as default, which embeds invisible attack

---

[1]Code is available at https://github.com/JiayinXu5499/PSIC.

patterns capable of significantly misleading the targeted VLP model; and the *full version*, which, with a customizable input condition (*e.g.*, keywords), provides the image's full semantic information. Moreover, to fulfill these mutually exclusive requirements within a single network, the proposed PSIC incorporates an adaptive multi-objective optimization strategy, where the model alternates between training under the rate-distortion and rate-encryption criteria, guided by the corresponding mode indicator. Besides using the intuitive optimization function that minimizes the similarity between the reconstructed image and its corresponding text counterpart (interfering with the fitting to certain labels), we further propose an Uncertainty-Aware Encryption-Oriented (UAEO) optimization function. The proposed UAEO automatically identifies uncertainty among the image-text pairs using the Dempster-Shafer Theory, and to maximize the similarity between cross-modal matches with higher matching uncertainty (forcing fitting to uncertain wrong labels) during the training process. The proposed PSIC scheme is plug-and-play and can be seamlessly integrated into most existing learned image compression (LIC) models. Extensive experiments demonstrate that the proposed scheme effectively conceals privacy information by misleading VLP-based applications, such as text–image retrieval and image captioning, as well as common computer vision tasks like image classification and facial attribute analysis, all while maintaining the same rate-distortion performance as baseline LIC models. We summarize the main contributions as follows:

- We propose a flexible image compression network that can adapt to user needs by controlling whether images can be accessed by downstream VLP models, all while maintaining good perceptual quality.

- We propose a Conditional Latent Trigger Generation (CLTG) module and an adaptive multi-objective optimization strategy to enable a single bitstream to be decoded into two versions with mutually exclusive objectives.

- We introduce an Uncertainty-Aware Encryption-Oriented (UAEO) Optimization Function that examines the uncertainty within the target model's prior knowledge, thereby enhancing robustness and increasing the success rate of misleading the target VLP model.

## 2. Related Work

### 2.1. Visual Language Pre-trained Models

VLP involves a new machine learning paradigm trained on large datasets of images and corresponding text, aiming to develop generalized models that understand content integrating both visual and linguistic information. Existing VLP

models can be mainly categorized into single-stream and dual-stream architectures (Du et al., 2022). The former (Li et al., 2019) leverages a single network, *e.g.,* transformer, to fuse both image and text features, facilitating downstream tasks. In contrast, the latter utilizes separate encoders for images and texts. A representative example is CLIP (Radford et al., 2021), which employed distinct image and text encoders to align embeddings through a contrastive learning approach, thereby learning a shared representation between images and texts. CLIP has demonstrated excellent performance in a variety of tasks such as zero-shot classification and image captioning. Recent approaches have integrated the strengths of both single-stream and dual-stream architectures to enhance performance. For instance, BLIP (Li et al., 2022) introduced a multimodal mixture of encoder-decoder architecture along with a captioning and filtering mechanism to tackle vision-language understanding and generation tasks. Building upon this, BLIP-2 (Li et al., 2023) incorporated a lightweight Querying Transformer (Q-Former) to align image and text features, thereby improving multimodal processing capabilities and computational efficiency. Further advancing this line of research, InstructBLIP (Dai et al., 2023) integrated an instruction-aware feature extraction method, enabling the model to derive specific representations from images based on varying instructions.

## 2.2. Learned Image Compression

In the past decade, significant progress has been made in the field of LIC. Ballé *et al.* (Ballé et al., 2017) were the pioneers in proposing the first end-to-end image compression network based on Variational Autoencoders (VAE) (Kingma & Welling, 2013). This was followed by the introduction of prior networks and context-based entropy models, which significantly improved the rate-distortion performance. Along this research direction, many efforts have been dedicated to further improving the entropy module. For example, Guo *et al.* (Guo et al., 2021) introduced enhanced entropy coding with global contextual information, while the chessboard context model and channel context were introduced in He *et al.* (He et al., 2021) and Minnen *et al.* (Minnen & Singh, 2020), respectively, to reduce the computational complexity of entropy encoding. Furthermore, Jiang *et al.* (Jiang et al., 2023) combined local spatial, global spatial, and channel context to propose a multi-reference entropy model, achieving state-of-the-art performance at the time. Beyond entropy models, other research works have focused on improving compression performance by adopting more powerful backbone networks. Notably, Rippel *et al.* (Rippel & Bourdev, 2017) proposed a GAN-based image encoder that improved reconstruction quality through adversarial loss. Recently, Yang *et al.* (Yang & Mandt, 2024) introduced an end-to-end optimized compression framework based on diffusion generative models, further advancing the development of lossy

image compression techniques.

## 2.3. Backdoor Attacks

Deep learning has been shown to be highly susceptible to backdoor attacks. These attacks typically involve modifying a model's parameters or input data, with the goal of poisoning training samples by injecting triggers to embed malicious patterns (Gao et al., 2023). When the model encounters a specific trigger during inference, it produces incorrect outputs. Due to the malicious nature of backdoor attacks, they have garnered significant attention across various domains, such as BadNet (Gu et al., 2017), SIG (Barni et al., 2019), and SSBA (Li et al., 2021). In the field of Vision-Language Pretraining (VLP), various strategies are employed for attacking purposes. For instance, targeting CLIP, Yang *et al.* (Yang et al., 2023) adjusted the encoder to maximize the cosine similarity between image and text embeddings, leading to misclassification in image-text retrieval tasks. Meanwhile, Bai *et al.* (Bai et al., 2024) utilized prompt learning techniques to train learnable parameters that distort the latent representations generated by the encoder. Liang *et al.* (Liang et al., 2024) proposed a dual-embedding guided framework for backdoor attacks, characterized by its stealthiness against backdoor detection due to the subtle parameter changes it induces.

While these attack methods have demonstrated significant effectiveness, they operate under the assumption that input images are pristine. However, in practical scenarios, images are typically compressed before being delivered to the target model. Overlooking the compression process can inevitably lead to unsatisfactory attack performance. In this context, Yu *et al.* (Yu et al., 2023; 2024) investigated how specific trigger patterns can be disrupted during the compression process, leading to compressed images that mislead downstream machine analytic tasks.

## 3. Privacy-Shielded Image Compression

### 3.1. Overview and Motivations

As mentioned above, our method Privacy-Shielded Image Compression (PSIC) is designed based on the critical motivation: supporting both visual perception and machine recognition, potentially under different encoding modes through coding and integrating these two types of information and activating them under specific conditions. In general, PSIC is characterized by its flexible conditional representation capacity, allowing a single bitstream to be decoded into two different versions: the *full* version, which preserves satisfactory perceptual quality and full semantic information, and the *encrypted* version, which contains nearly invisible adversarial attack patterns that significantly mislead downstream VLP or machine vision models. More-

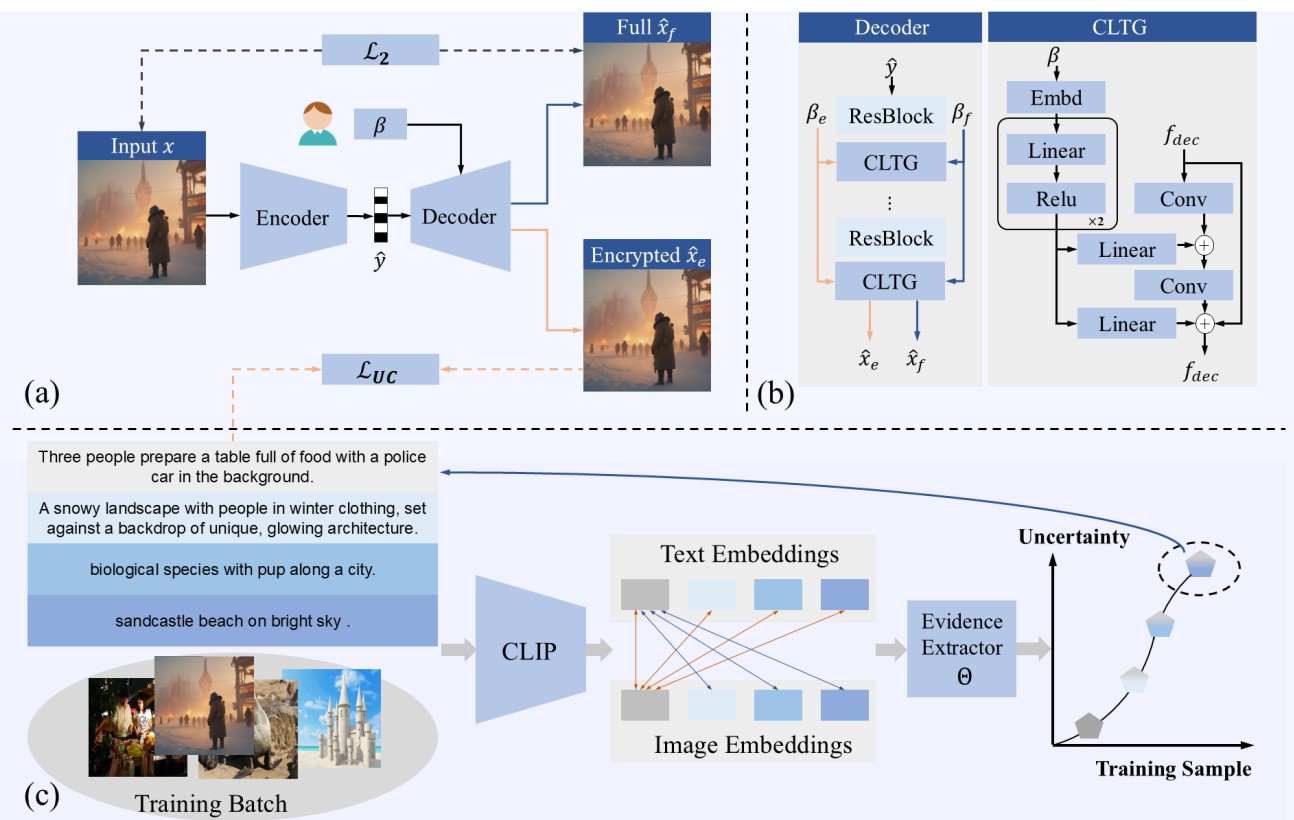

*Figure 2.* (a) Compressing pipeline of the proposed Privacy-Shielded Image Compression (PSIC). (b) Details of the proposed Conditional Latent trigger Generation (CLTG) module. (c) Intuitive illustration of the introduced Uncertainty-Aware Encryption-Oriented (UAEO) optimization function.

over, under the default setting, only the *encrypted* version is produced. The *full* version is generated only when users provide customizable conditions (*e.g.*, keywords) to the decoding side.

The overall framework is illustrated in Fig. 2 (a). At the encoding side, the input image $x$ would be first fed to an encoder $E(\cdot)$ to obtain its compact latent representation $\hat{y}$,

$$\hat{y} = E(x). \tag{1}$$

An entropy model $Q(\cdot)$ is leveraged to produce the binary bitstream of $\hat{y}$ and provide its bitrate indices $r$. Moreover, alongside the bitstream, the customizable condition $\beta$ must also be provided by the users and transmitted to the decoding side. For clarity, we denote the $\beta_e$ and $\beta_f$ as the indicators for the *encrypted* version and *full* version, respectively.

At the decoding side, the $\beta \in [\beta_e, \beta_f]$ would be fed to a trigger generator $T(\cdot)$, obtaining corresponding feature-wise triggers $\tau \in [\tau_e, \tau_f]$,

$$\tau_e = T(\beta_e), \ \ \tau_f = T(\beta_f). \tag{2}$$

The triggers would be injected to the latent representation $\hat{y}$ and jointly fed to the decoder $D(\cdot)$, generating the corresponding version tailored to the user's requirements,

$$\hat{x}_e = D(\hat{y}, \tau_e), \ \ \hat{x}_f = D(\hat{y}, \tau_f), \tag{3}$$

where $\hat{x}_e$ and $\hat{x}_f$ denote the *encrypted* version and *full* version, respectively.

In realizing the challenging coding pipeline mentioned above with satisfactory compression efficiency, three innovative designs play key roles: *1)* the ***Conditional Latent Trigger Generation (CLTG) module***, which enables the generation of different triggers through a single module, influences the decoding process to achieve mutually exclusive objectives; *2)* the ***Uncertainty-Aware Encryption-Oriented (UAEO) optimization function***, which leverages the CLIP model as the attacking target, being capable of automatically identifing low-confidence cross-modal matches in the training set via Dempster-Shafer Theory, enhancing robustness and the success rate of attacks; *3)* the ***adaptive multi-objective optimization strategy*** automatically balances the distinct requirements of *encrypted* and *full* version reconstruction, instead of relying on an empirically decided approach, leading to improved compression performance. In

the following subsections, the CLTG module, the UAEO optimization function, and the adaptive multi-objective optimization strategy will be detailed in Subsec. 3.2, Subsec. 3.3 and Subsec. 3.4, respectively.

### 3.2. Conditional Latent Trigger Generation Module

Compared to existing LIC-oriented backdoor attack approaches, where the trigger generation process considers only the attacking purpose, PSIC faces a more significant challenge: it aims to reconstruct a single bitstream into two versions with mutually exclusive objectives. To address this, we propose a progressive trigger injection strategy that deploys a series of CLTG modules with independent parameters at each decoding block, as shown in Fig. 2 (b). The deployed CLTG modules take the mode indicator $\beta \in \{\beta_e, \beta_f\}$ as input and adaptively produce bias information at different representation stages.

In particular, within the CLTG module, the mode indicator $\beta$ would be first fed in to a 2-layer MLP, obtaining the corresponding conditional bias feature $f_\beta$. This bias feature, $f_\beta$, is then fused with the image feature to bias the decoding process towards its indicated preference,

$$f_{CLTG} = f_\beta W_d + f_{dec}, \qquad (4)$$

where $f_{CLTG}$ and $f_{dec}$ denote the output and the input features of the CLTG module respectively, $W_d$ is the feature element-wise weighting map with $d$ denoting the channel depth aligned with the input features.

### 3.3. Uncertainty-Aware Encryption-Oriented (UAEO) Optimization Function

In the proposed PSIC, the decoded encrypted version is expected to contain adversarial patterns that significantly mislead the target CLIP model. To this end, a simple strategy involves setting an objective function for the *encrypted* version $\hat{x}_e$ reconstruction by minimizing the cosine similarity between $\hat{x}_e$ and its textual prompt $t$ in the CLIP feature space. However, this approach is inevitably not capable of providing a stable and efficient optimization path, as it does not identify a specific target during the training process. Moreover, acknowledging that the existing multimodal training data is often quite raw, with many datasets collected from the Internet, it inevitably contains image pairs that are not well aligned. Directly applying the naive objective function may result in issues with robustness and efficiency. Given the above considerations, we are motivated to capture and leverage the uncertainty within the CLIP model's prior knowledge, as illustrated in Fig. 2 (c). In general, the *encryption*-oriented optimization involves in maximizing the similarity between cross-modal matches with higher matching uncertainty, while decreasing the similarity with the rest within a training batch. The Dempster-Shafer Theory (DST)

(Dempster, 2008) of evidence, which combines evidence from different sources with a degree of belief (also known as the belief function), is adapted to obtain pair uncertainty.

To be specific, consider the input image $x_i$, and text $t_j$ within a batch of size $K$, *i.e.*, $i, j \in [1, K]$, where $i = j$ indicates they are labeled as a pair. For a certain image-text pairs $(x_i, t_j)$, we employ the evidence extractor proposed in (Qin et al., 2022) $\Theta(\cdot)$ to obtain their evidence $e_{ij}$,

$$e_{ij} = \Theta(x_i, t_j) = exp^{\tanh(\text{Clip}(x_i, t_j)/s)}, \qquad (5)$$

where $\text{Clip}(\cdot, \cdot)$ denotes the cosine similarly in the CLIP feature space, $s \in (0, 1)$ is a scaling factor.

Accordingly, the image-to-test evidence vector within a batch $e_i^{i2t} = [e_{i1}, e_{i2}, ..., e_{iK}]$ would be obtained, as well as the test-to-image evidence vector $e_j^{t2i} = [e_{1j}, e_{2j}, ..., e_{Kj}]$. Then, we calculate the cross-modal bidirectional evidence vector via,

$$e_i^b = e_i^{i2t} + e_i^{t2i} = [e_{i1}^b, ..., e_{iK}^b]. \qquad (6)$$

Thus, the corresponding Dirichlet distribution strength is parameterized by $[e_{i1}^b + 1, e_{i2}^b + 1, ..., e_{iK}^b + 1]$ with the distribution strength approximated by $\sum_{j=1}^{K}(e_{ij}^b + 1)$. And the uncertainty mass $u_{ij}$ of the image-text pair $(x_i, t_j)$ can be calculated as

$$u_{ij} = \frac{e_{ij}^b}{\sum_{j=1}^{K}(e_{ij}^b + 1)}. \qquad (7)$$

Subsequently, for the $i$-th image $x_i$, the corresponding prompt $t_n$ with highest uncertainty could be obtained via,

$$t_n = \arg\min_n u_{in}, \qquad (8)$$

Thus, the encryption-oriented optimization function $\mathcal{L}_{UC}$ can be formulated by,

$$\mathcal{L}_{UC}(x_i) = -log \frac{exp(\text{CLIP}(f(x_i), g(t_n))/s)}{\sum_{j=1}^{K} exp(\text{CLIP}(f(x_i), g(t_j))/s)}. \qquad (9)$$

### 3.4. The Adaptive Multi-Objective Optimization Strategy

The proposed PSIC is designed to obtain a *compact* latent representation from the encoding stage, which is then adaptively decoded into two versions that meet the *encryption* and *perception* requirements, respectively. Given the significant challenges posed by the mutual exclusivity of the three objectives, we develop a two-stage training strategy. The first stage focuses on *compactness*, while the second stage addresses the divergent representations tailored for *encryption* and *perception* requirements.

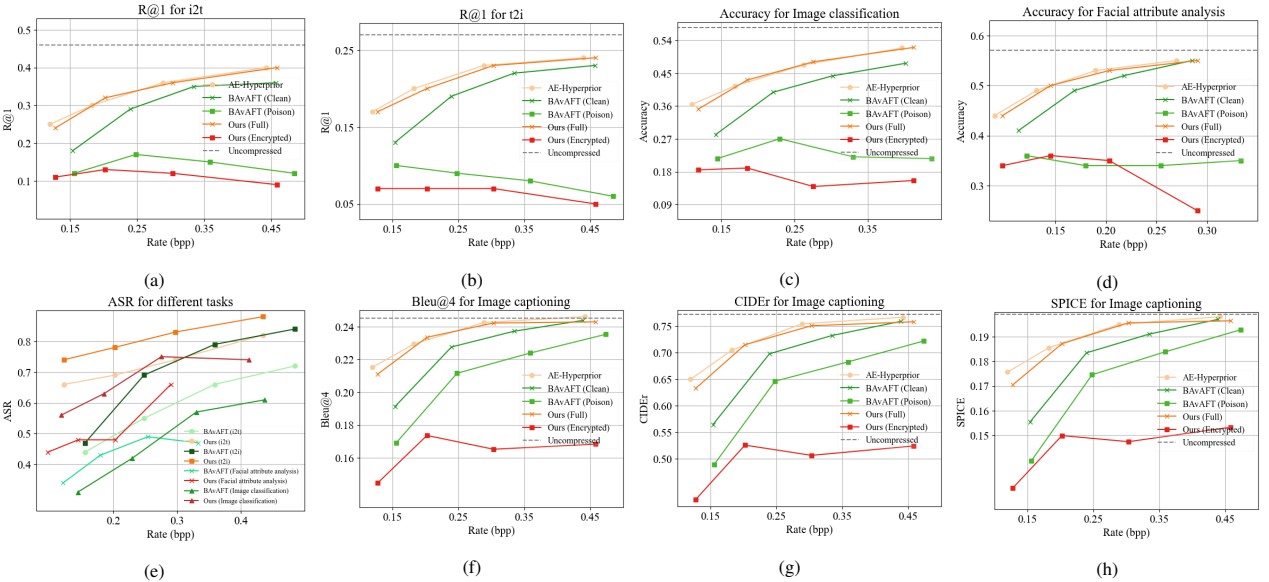

*Figure 3.* Performance comparisons regarding the four employed downstream tasks.

*First Stage.* As for the first stage, the encoder $E_\theta$, entropy model $Q_\phi$, and the conditional decoder $D_\psi$, parameterized by $\theta$, $\phi$, $\psi$, respectively, are jointly optimized by alternating between *rate-encryption* and *rate-perception* criteria. In particular, we split the training iterations within one epoch into two sessions, and the corresponding training process can be formulated by:

$$S\ 1\text{-}1{:}\bar{\theta}, \bar{\phi}, \bar{\psi} = \underset{(\phi,\psi,\theta)}{\arg\min} \sum_{(x \in X)} \lambda \cdot \mathcal{L}_2(x, D_\psi(E_\theta(x), \beta_f))$$
$$+ Q_\phi(\hat{y}),$$

$$S\ 1\text{-}2{:}\tilde{\theta}, \tilde{\phi}, \tilde{\psi} = \underset{(\bar{\theta},\bar{\phi},\bar{\psi})}{\arg\min} \sum_{(x \in X)} \lambda \cdot \big( \mathcal{L}_2(x, D_{\bar{\psi}}(E_{\bar{\theta}}(x), \beta_e))$$
$$+ \mathcal{L}_{UC}(D_{\bar{\psi}}(E_{\bar{\theta}}(x), \beta_e))) + Q_{\bar{\phi}}(\hat{y}),$$

where $\lambda$ is a Lagrange parameter to obtain the ICM models at different compression levels.

*Second Stage.* In the second stage, the parameters of the encoder and entropy model are frozen, allowing the focus to shift towards the divergent representations needed to fulfill the *encryption* and *perception* requirements. Specifically, the iterations within each epoch are alternately optimized with the following criteria,

$$S\ 2\text{-}1{:}\bar{\psi} = \underset{(\psi)}{\arg\min} \sum_{(x \in X)} \mathcal{L}_2(x, D_\psi(E_\theta(x), \beta_f)),$$

$$S\ 2\text{-}2{:}\tilde{\psi} = \underset{(\bar{\psi})}{\arg\min} \sum_{(x \in X)} \mathcal{L}_{UC}(D_{\bar{\psi}}(E_\theta(x), \beta_e)).$$

## 4. Experiments

During our experiments, we compare LIC networks both with and without our PSIC implementation to assess perceptual quality and encryption effectiveness. Additionally, we employ a cutting-edge LIC-oriented backdoor attack method to demonstrate the superiority of the proposed PSIC. Furthermore, we conduct comprehensive ablation studies to highlight the effectiveness of our specially designed modules.

### 4.1. Experimental Setup

***Benchmark.*** Four downstream tasks are employed for comprehensive evaluation, including two cross-modal applications (image-text retrieval and image captioning) and two common machine vision tasks (image classification and facial attribute analysis).

In particular, image-text retrieval, image classification, and facial attribute analysis are performed directly by the target model, CLIP (ViT-B/32), during training. Meanwhile, image captioning is conducted on the unseen BLIP-2 (OPT-2.7b) (Li et al., 2023), to demonstrate the generalization capacity.

- *Image-text retrieval:* Performances for both the image-to-text (i2t) and text-to-image (t2i) tasks are evaluated using Recall@1 (R@1).

- *Image classification:* We employ 5,000 images from the ILSVRC 2012 dataset (Deng et al., 2009) (ImageNet-1k). Performance is assessed using Top-1 accuracy.

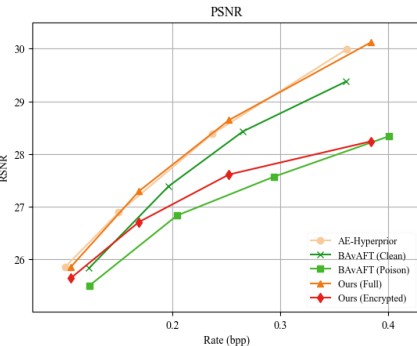

*Figure 4.* Perceptual quality comparisons in terms of PSNR among the proposed PSIC, the LIC backbone network, and BAvAFT.

- *Facial attribute analysis:* We utilize 10,000 images from the CelebA dataset (Liu et al., 2018), each labeled with 40 attributes (e.g., gender, age, expression). Performance is evaluated using Top-1 accuracy.

- *Image captioning:* The entire validation set of Flickr8k dataset (Hodosh et al., 2013) is employed. The performance is indexed by BLEU@4, CIDEr, and SPICE. In particular, BLEU@n analyzes the proportion of n-grams in the candidate translation that appear in the reference translation. CIDEr primarily assesses the quality of image captions by calculating the similarity between the generated description and a set of reference descriptions. SPICE, on the other hand, focuses on semantic accuracy by using scene graphs to compare objects, attributes, and relationships in the captions.

- *Human perception:* The Kodak dataset (Kodak, 1993) is utilized to evaluate perceptual quality, with PSNR serving as the assessment metric.

To intuitively demonstrate the effectiveness of misleading downstream tasks, we additionally adopt the Attack Success Rate (ASR) metric for image-text retrieval, image classification, and facial attribute analysis. In particular, the ASR is defined as the percentage of samples that are misclassified due to adversarial attacks. In our context, ASR measures the proportion of samples from the baseline Learned Image Compression (LIC) model that are correctly processed in downstream tasks but are misled when processed through our proposed model.

***Baseline.*** Regarding the backbone LIC network, the milestone *AE-Hyperprior* (Ballé et al., 2018) is adopted. Specifically, both the proposed PSIC and the backbone network are trained by randomly selected 70,000 image-text pairs from the CC3M dataset (Sharma et al., 2018) from scratch for a fair comparison. Moreover, a cutting-edge LIC-oriented backdoor attack method, BAvAFT, is also employed for

comparison. Specifically, BAvAFT involves injecting triggers into the input images with the expectation that the reconstructed images will significantly mislead downstream models. We implemented the BAvAFT on the same backbone LIC network, training it from scratch using the same training set and targeting the Vision Transformer model ViT-B/32, consistent with our PSIC.

***Implementation details.*** In the first stage of the training process, we use the Adam optimizer with a learning rate of 1e-4 and train for 200 epochs, which will be reduced to 1e-5 and train for an additional 100 epochs for the second stage. Moreover, all models are trained with a batch size of 128, implemented in PyTorch with CUDA support, and trained on a single NVIDIA A8000-80G GPU. They are all trained four times with different Lagrange parameters, obtaining four distinct compression levels regarding bpp.

### 4.2. Experimental Results

***Encryption efficiency.*** The encryption efficiency comparisons for image-text retrieval, image classification, and facial attribute analysis are shown in Fig. 3 (a)-(e). Encouraging results have been observed, showing that the *encrypted* version proposed by PSIC can significantly mislead the target CLIP model, while the *full* version effectively preserves the full semantic information. In particular, compared to the backbone LIC network, our *encrypted* version achieves an average ASR of 80.8%, 72.3%, 67.0%, 51.5% for text-to-image, image-to-text retrieval, image classification, facial attribute analysis, respectively. Meanwhile, the *full* version maintains comparable machine analytics performance to the AE-Hyperprior. Considering both versions are reconstructed from the same bitstream and exhibit mutually exclusive semantic representations, these results are remarkable, owing to our multistage trigger injection and adaptive multi-objective optimization strategies.

Meanwhile, compared to the cutting-edge backdoor attack method BAvAFT, the proposed PSIC offers overwhelming advantages in encryption (attacking) efficiency, achieving an average ASR improvement of 12.9% regarding all of the downstream tasks. Additionally, our *full* version consistently outperforms the *clean* version across multiple tasks and bpp points. According to our analysis, the main advancement lies in injecting the triggers at the representation stage, rather than the preprocessing stage, as this approach allows the optimized encoder to better capture the key information critical for both attacking and normal purposes. Moreover, the performance on the unseen image captioning task in Fig. 3 (f), (g), and (h) further demonstrates the effectiveness and superiority of our method over BAvAFT, highlighting the robustness and generalization capability of the PSIC.

***Perceptual Quality.*** Perceptual quality comparisons in

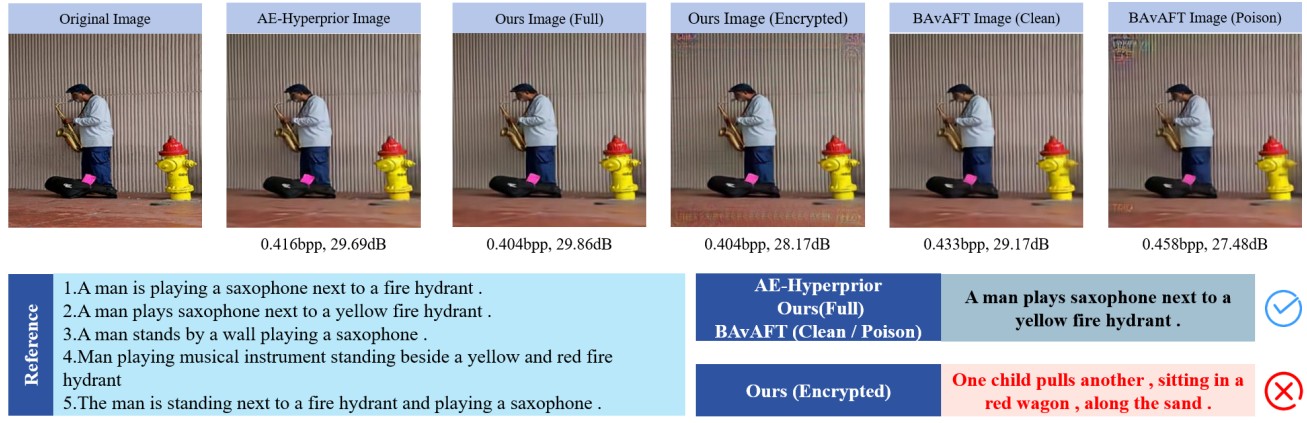

*Figure 5.* Visualization of performance in terms of perceptual quality and encryption efficiency.

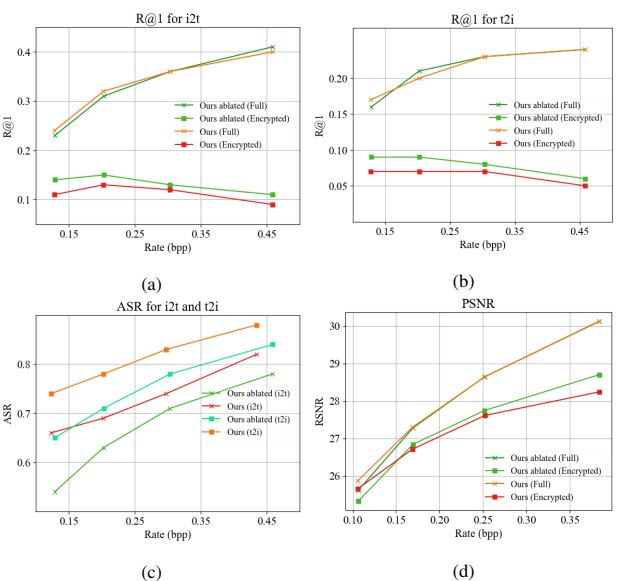

*Figure 6.* Ablation results for the proposed UAEO optimization function, illustrating encryption efficiency in (a), (b), and (c), and perceptual quality in (d).

terms of PSNR is provided in Fig. 4. As shown, our *full* version maintains the same *rate-perception* level as the backbone LIC, while also showing an improvement over the *clean* mode of BAvAFT, indicated by an average increase of 1.0 dB across the four bpp points. Moreover, considering that BAvAFT requires separate encoding processes for *clean* and *poison* modes, the improvement is remarkable. Meanwhile, our *encrypted* version also maintains favorable perceptual quality, demonstrating its capacity to meet common perception-oriented requirements. To provide insight into the overall performance, a group of visualized examples is shown in Fig. 5.

### 4.3. Ablation Study

In this subsection, we perform an ablation study on the UAEO optimization function to demonstrate its effectiveness. In this ablation, we use a naive attacking objective function—i.e., minimizing the cosine similarity between the reconstructed encrypted version and its corresponding text—as the ablated version. Comparison results between our full version and the ablated version at the same compression level are provided in Fig. 6. As shown, the uncertainty-aware strategy significantly boosts encryption efficiency, resulting in an average ASR improvement of 6.3% and 6.2% for text-to-image and image-to-text retrieval tasks, respectively, thereby illustrating its effectiveness in capturing and leveraging uncertainty within the training data.

## 5. Conclusions

This paper proposes a novel PSIC scheme aimed at protecting user privacy from exploitation by VLP models at the image compression stage. PSIC is characterized by its ability to produce bitstreams with two optional decoding versions: an *encrypted* version that offers satisfactory perceptual quality yet remains inaccessible to VLP models, and a *full* version providing complete semantic information. These capabilities are achieved through two key components. First, a CLTG module generates bias information based on customizable conditions, guiding the decoder toward different reconstructed outputs. Second, an UAEO optimization function leverages the target VLP model's uncertainty on the training data to provide an efficient and robust optimization strategy. Moreover, an adaptive multi-objective optimization approach is explored to simultaneously enhance encryption performance and perceptual quality within a unified training process. Experimental results and ablation studies validate the effectiveness of our design.

## Impact Statement

As deep learning models increasingly rely on publicly available visual data, ensuring privacy-preserving mechanisms becomes crucial for preventing unintended data exploitation. This paper introduces a privacy-shielding image compression (PSIC) framework, aiming to protect image data from unauthorized use by Vision-Language Pretrained (VLP) models and general machine vision analytics. By introducing multiple decoding options, it enables individuals and organizations to maintain control over their visual data while still benefiting from advanced AI technologies. Ethically, this approach promotes data ownership and responsible AI usage, reducing the risk of unauthorized surveillance or unintended data exploitation. However, similar techniques could also be leveraged to potentially negative applications, including hindering lawful model training. We believe that our work contributes to ongoing discussions about ethical AI and responsible data usage, highlighting the importance of developing privacy-preserving techniques that balance protection with ethical considerations. This, in turn, should inspire further research toward more robust, fair, and trustworthy AI-driven data protection strategies.

## Acknowledgements

This work was supported by the Guangdong Basic and Applied Basic Research Foundation under Grant 2024A1515010454.

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
