# OpenReview forum: "Privacy-Shielded Image Compression: Defending Against Exploitation from Vision-Language Pretrained Models"
_ICML.cc/2025/Conference — ICML 2025 poster_

### Official Review · Reviewer_BN9M · 2025-03-11

**Overall Recommendation:** 3

**Summary:**

This paper proposes a novel Privacy-Shielded Image Compression (PSIC) method aimed at protecting images from being exploited by Vision-Language Pretrained (VLP) models. The PSIC framework integrates an adaptive multi-objective optimization strategy that balances perceptual quality and encryption effectiveness. A Conditional Latent Trigger Generation (CLTG) module is introduced to generate different decoding options from a single bitstream, while the Uncertainty-Aware Encryption-Oriented (UAEO) optimization function is used to maximize encryption efficiency against VLP models. Experimental results indicate that PSIC significantly degrades the interpretability of compressed images by VLP models while maintaining visual quality for human perception.

**Claims And Evidence:**

The claim: "The proposed PSIC scheme is plug-and-play and can be seamlessly integrated into most existing learned image compression (LIC) models" is not well supported. The paper only evaluated PSIC on one backbone (AE-Hyperprior, 2018). How about more recent LIC models such as HiFiC[1], ELIC[2] and MLIC[3]? If the experiments on these more recent models are not possibly all provided, please clarify what cost it will take to equip the general LIC model with the proposed PSIC method, to support that "the proposed PSIC scheme is plug-and-play."

[1] High-fidelity generative image compression[C]. Advances in neural information processing systems, 2020, 33: 11913-11924.

[2] ELIC: Efficient learned image compression with unevenly grouped space-channel contextual adaptive coding[C]//Proceedings of the IEEE/CVF Conference on Computer Vision and Pattern Recognition. 2022: 5718-5727.

[3] MLIC: Multi-reference entropy model for learned image compression[C]//Proceedings of the 31st ACM International Conference on Multimedia. 2023: 7618-7627.

**Essential References Not Discussed:**

BAvAFT mentioned in the experiment section is not cited, and it is unclear if it is a fair baseline.

**Experimental Designs Or Analyses:**

1. The ablation study in the experiment part is useful but would be better to extend to analyze other proposed modules or strategies, such as the proposed adaptive multi-objective optimization.
2. In perceptual quality comparison, instead of comparing to AE-Hyperprior (2018), more recent general LIC models that optimized for perceptual quality should be included to provide baselines. Additionally, more quality metrics should be evaluated, such as those that are more aligned with the human vision system: SSIM, LPIPS, NIQE, CLIPIQA.
3. The paper evaluates the method on four downstream tasks: image-text retrieval, image classification, facial attribute analysis, and image captioning. These are reasonable benchmarks, but additional comparison with recent works in Yu et al. 2023, 2024 (which is reviewed in related work) is necessary, since these works are also designed for misleading downstream machine analytic tasks.

**Methods And Evaluation Criteria:**

The method is compared against BAvAFT, but the baseline method lacks proper citation. A clear description of BAvAFT should be provided.

**Other Comments Or Suggestions:**

In addition to the one row of images in Fig. 5, please show more visual comparison results to demonstrate the effectiveness of the proposed method (both full and encrypted modes) on perceptual quality.

**Other Strengths And Weaknesses:**

Strengths:
1. The proposed Privacy-Shielded Image Compression (PSIC) framework introduces a novel approach to integrating privacy protection directly into the image compression process, which is an underexplored area in learned image compression.
2. The idea of conditional latent trigger generation (CLTG) to enable different decoding versions from a single bitstream is innovative and provides flexibility in balancing privacy and perceptual quality.
3. The method is designed to be plug-and-play, potentially making it applicable to a broad range of learned image compression (LIC) models.

Weaknesses:

My identified weaknesses have been thoroughly mentioned in other parts of the review, so no additional weaknesses will be mentioned here.

**Questions For Authors:**

1. Why does the experimental backbone use AE-Hyperprior (2018)? Have you considered evaluating PSIC on newer LIC models?
2. Why is there no direct comparison with Yu et al. (2023, 2024), given that their work also targets misleading downstream machine analysis tasks?

**Relation To Broader Scientific Literature:**

Previous related works are either focusing on misleading downstream machine analytic tasks or designing pre-processing methods on the input data. The paper focuses on privacy shielding against exploitation from vision-language pretrained models, and it addresses this problem from the aspect of the compression phase, which is interesting and novel.

**Theoretical Claims:**

The Dempster-Shafer Theory used to model pair uncertainty is presented as a black-box implementation with minimal discussion. More discussion would be helpful to understand.

---

> ### Author Rebuttal · Authors · 2025-04-01
>
> First of all, we greatly appreciate your thorough review and helpful suggestions. Below, we address your concerns one by one, and we hope our responses fully clarify each point. If any misunderstanding remains, we sincerely welcome further clarification or suggestions.
>
> Q.1 Incorporate more quality metrics.
>
> Ans.: Thanks for your suggestion. We have further evaluated the perceptual quality using additional metrics, including SSIM, LPIPS, NIQE, and CLIPIQA. The results corresponding to Fig. 4 of our manuscript can be found at the following link. https://pic1.imgdb.cn/item/67eb75140ba3d5a1d7e8f36d.png It should be noted that the NIQE curves appear somewhat irregular and produce unusual results compared to the other four metrics. Given that NIQE is a no-reference IQA method with known instability in certain scenarios,  it may not accurately reflect the actual perceptual quality in this case.
>
> Q.2: Perform the PSIC scheme on other LIC models.
>
> Ans.: We appreciate your insightful suggestion. We have implemented the PSIC in two cutting-edge LIC models (ELIC, and MLIC++). To ensure a fair comparison, all training settings strictly follow those described in our original manuscript. The corresponding results can be found in below table.
>
> ||Meth.|PSNR|i2t|t2i|ASR(i2t)|ASR(t2i)|
> |-|-|-|-|-|-|-|
> |ELIC(bpp=0.19)|Baseline|27.24|0.32|0.22|-|-|
> ||Ours($x_e$)|26.64|0.30|0.19|-|-|
> ||Ours($x_f$)|26.05|0.12|0.07|0.74|0.79|
> |MLIC++(bpp=0.18)|Baseline|27.26|0.35|0.22|-|-|
> ||Ours($x_e$)|26.93|0.32|0.21|-|-|
> ||Ours($x_f$)|26.44|0.13|0.09|0.69|0.73|
>
> These results demonstrate the effectiveness of the PSIC method on other LIC models, as it achieves perceptual quality comparable to the baseline model while offering remarkable encryption efficiency in terms of ASR.  Due to time constraints, we will include more comprehensive results on additional LIC models across multiple BPP points in the supplementary material of our revised paper. We appreciate your understanding.
>
> Q.3 Lacking citation of BAvAFT and compare with recent works (Yu et al. 2023, 2024).
>
> Ans.: We apologize for the confusion caused by the missing citation for BAvAFT. In fact, BAvAFT originates from the papers you mentioned (Yu et al., 2023, 2024. The 2024 version is an extension of the 2023 version, and we employed the latest version in our manuscript). We will properly cite both works in the revised version to avoid confusion and to give appropriate confirmation.
>
> Q.4. Ablation study on CLTG module.
>
> Ans.: We provide an ablation study on CLTG. We remove all CLTG modules and train only the compression backbone using UAEO-based loss function:$λ_1 L_2(x, \widehat{x}) + λ_2 L_{UC} (\widehat{x}) + r$, $\widehat{x}$ is the reconstructed image, $r$ is the bitrate. We performed the comparison at a BPP level of 0.20, and the results are provided below.
>
> |Bpp|Meth.|PSNR|i2t|ASR(i2t)|t2i|ASR(t2i)|
> |-|-|-|-|-|-|-|
> |0.20|Ours($x_f$)|26.07|0.32|-|0.20|-|
> ||Ours($x_e$)|25.61|0.13|0.69|0.07|0.78|
> ||*w/o* CLTG|24.48|0.11|0.73|0.08|0.75|
>
> As shown, removing CLTG leads to a PSNR drop of 1.59 dB at the same ASR level.
>
> Q.5. Ablation study on multi-stage training strategy.
>
> Ans.: We further conduct an additional ablation study on the proposed multi-stage training strategy. In particular, we omit Stage 2 and instead train the model using the settings from Stage 1 until convergence (an additional 100 epochs). We present the comparison results regarding encryption performance (encrypted version) and perceptual quality (full version) below.
>
> |Bpp|Meth.|ASR(i2t)|ASR(t2i)|PSNR($x_f$/$x_e$)|
> |-|-|-|-|-|
> |0.13|Ours|0.66|0.74|24.72/24.57|
> ||*w/o* stg2|0.05|0.06|24.70/24.68|
> |0.20|Ours|0.69|0.78|26.07/25.61|
> ||*w/o* stg2|0.03|0.05|26.02/26.02|
> |0.30|Ours|0.74|0.83|27.37/25.67|
> ||*w/o* stg2|0.04|0.06|27.28/27.27|
> |0.46|Ours|0.82|0.88|28.78/27.34|
> ||*w/o* stg2|0.02|0.04|28.84/28.84|
>
> The effectiveness of the multi-stage training strategy is readily apparent, as it improves encryption performance by about 72% in terms of the average ASR.
>
> Q.6: Provide more visual comparisons
>
> Ans.: Thanks for your kind suggestions. We have provided four extra groups of visual compression in addition to Fig. 5, which can be observed in the following links. https://pic1.imgdb.cn/item/67eb9c9c0ba3d5a1d7e9030c.jpg
>
> Q.7 Provide a supplemental material.
>
> Ans.: We appreciate this constructive suggestion and will incorporate the following in the supplementary material:
>
> a) Extended ablation studies on the CLTG module and the multi-stage training strategy across multiple BPP points;
>
> b) Comprehensive evaluations demonstrating the implementation of PSIC on other baseline LIC models, along with multiple perceptual quality metrics;
>
> c) Additional visual examples generated by the PSIC method, the compression baseline, and the employed BAvAFT method in the supplementary material;
>
> d) A detailed introduction to the Dempster-Shafer Theory, along with the derivation process of the employed evidence extractor.

---

> > ### Comment · Reviewer_BN9M · 2025-04-02
> >
> > Thanks for the author's response. I have several concerns as follows:
> >
> > 1. Could you give more analysis or discussion regarding the performance comparison on different quality metrics? And your claim "NIQE is a no-reference IQA method with known instability in certain scenarios" needs more clarification, since as far as I know, many works evaluate their methods on NIQE for perceptual quality measurement.
> >
> > 2. The output images encrypted by the proposed method often remain some artifacts on the top and bottom areas. Could you give some analysis or discussion on this artifact? Given that the poison images from BAvAFT show smaller artifact area (just top left), could this visual results be viewed as drawbacks of the proposed method regarding visual performance?

---

> > > ### Author Response · Authors · 2025-04-03
> > >
> > > Q1.1 Could you give more analysis or discussion regarding the performance comparison on different quality metrics?
> > >
> > > Ans. : Thanks for your suggestions. First, the performance curves of PSNR, SSIM, LPIPS, and CLIPIQA exhibit a consistent trend, indicating that our *full* version closely matches the compression baseline in terms of perceptual quality. In particular, for PSNR, SSIM, and LPIPS, our *full* version’s performance is almost identical to that of the compression baseline. Moreover, it demonstrates a notable improvement over the *clean* version of BAvAFT, with average gains of 0.224, 0.009, 0.024, and 0.025 for PSNR, SSIM, LPIPS, and CLIPIQA, respectively. These results highlight the effectiveness of our PSIC method in providing a perceptually satisfactory compression pipeline.
> > >
> > > Q1.2 And your claim "NIQE is a no-reference IQA method with known instability in certain scenarios" needs more clarification, since as far as I know, many works evaluate their methods on NIQE for perceptual quality measurement.
> > >
> > > Ans. : Thank you for pointing this out, and we apologize for our previous wording. NIQE is indeed a widely adopted and milestone perceptual QA metric with proven effectiveness in various applications, *e.g.*, image compression and restoration. In particular, NIQE evaluates perceptual quality by measuring the distance (*e.g.*, Mahalanobis distance) between the feature distribution of a test image and that of the natural images, based on natural scene statistics (NSS).
> > >
> > > However, in our experiments, the NIQE curves show counter-intuitive trends: our encrypted version shows significantly better scores over the compression baseline, while the poisoned version of BAvAFT also outperforms its clean counterpart. These results are not aligned with those of other metrics (*e.g.*, LPIPS, CLIP-IQA) and do not reflect actual visual perception.
> > >
> > > Through in-depth analysis, we attribute these irregularities to two main factors: 1) Region Selection Bias: NIQE selects sharp regions in the test images and uses their features in the assessment process. However, in our PSIC and BAvAFT methods, the adversarial patterns often appear at the image borders or in less textured regions, which may be overlooked by NIQE’s sampling process. 2) Non-standard Distortion Types: The distortion patterns introduced by our PSIC method and the LIC-oriented backdoor attack method differ from those commonly found in standard IQA datasets (*e.g.*, Gaussian noise, quantization noise, or contrast degradation). Since NIQE’s model is not trained on these unfamiliar distortions, it may not accurately evaluate them, thereby producing unreliable scores in such cases.
> > >
> > > In summary, while NIQE remains a valuable metric, we advise caution when interpreting its results under adversarial or task-specific distortions. In our case, other perceptual metrics (*e.g.*, LPIPS, CLIP-IQA) offer more consistent and reliable assessments.
> > >
> > > Q.2 The output images encrypted by the proposed method often remain some artifacts on the top and bottom areas. Could you give some analysis or discussion on this artifact? Given that the poison images from BAvAFT show smaller artifact area (just top left), could this visual results be viewed as drawbacks of the proposed method regarding visual performance?
> > >
> > > Ans.: Thanks for your suggestions. Regarding the concern about increased artifacts in our encrypted version, we acknowledge that our encrypted images may exhibit more artifacts near the top and bottom regions. This is primarily due to the inherent trade-off between perceptual quality and encryption efficiency (ASR). Raising encryption efficiency inevitably comes at the expense of perceptual quality, as these two objectives are inherently conflicting. Thus, we believe that the presence of such artifacts alone should not be the sole criterion for evaluating visual performance; rather, a fair assessment requires comparisons under matched conditions and controlled variables.
> > >
> > > To provide a fair comparison, we trained a version of PSIC that achieves the same ASR as BAvAFT. A set of visual comparisons between our encrypted version and BAvAFT’s poison version can be found in the following link. https://pic1.imgdb.cn/item/67ed50a80ba3d5a1d7eb28bf.jpg
> > > Visual comparisons show that under matched encryption efficiency, both methods demonstrate comparable perceptual quality. While the artifact regions in PSIC may appear slightly more spatially distributed, they remain visually unobtrusive and acceptable in practical scenarios.
> > >
> > > Furthermore, we would like to emphasize a key advantage of our proposed PSIC: it enables both full and encrypted versions to be decoded from a single bitstream, without requiring any additional encoding steps. In contrast, BAvAFT needs an extra encoding process and two SEPARATE bitstreams—one each for the clean and poisoned versions. This design makes PSIC significantly more efficient and user-friendly, especially when balancing encryption needs and perceptual quality.

---

### Official Review · Reviewer_uJHB · 2025-03-12

**Overall Recommendation:** 3

**Summary:**

The paper presents a novel approach for privacy protection in image compression, termed Privacy-Shielded Image Compression (PSIC), aimed at defending against exploitation by Vision-Language Pretrained (VLP) models. The method leverages a flexible compression scheme that creates bitstreams with multiple decoding options. By default, the bitstream preserves perceptual quality while concealing semantic content from VLP models. The method can be adjusted to allow reconstruction of images with full semantic information when required. The system incorporates a Conditional Latent Trigger Generation (CLTG) module to produce bias information for guiding the decoding process and an Uncertainty-Aware Encryption-Oriented (UAEO) optimization function to maximize encryption performance while maintaining perceptual quality. The paper claims that PSIC can mislead VLP models and prevent them from exploiting compressed images for downstream tasks such as image-text retrieval and classification, while preserving image quality.

**Claims And Evidence:**

Yes, clear and convincing.

**Essential References Not Discussed:**

N/A.

**Experimental Designs Or Analyses:**

The experiments cover a range of downstream tasks (e.g., image classification, image-text retrieval, and facial attribute analysis) and provide solid evidence that the PSIC method outperforms existing approaches. The use of attack success rate (ASR) to measure the model's effectiveness in misleading VLP models and its ability to retain perceptual quality is appropriate. The ablation studies further validate the contributions of specific modules, such as the UAEO function.

**Methods And Evaluation Criteria:**

Yes.

**Other Comments Or Suggestions:**

N/A.

**Other Strengths And Weaknesses:**

The approach is innovative, particularly in the integration of image compression with privacy-preserving mechanisms. The ability to generate multiple versions of an image from the same bitstream—one that protects privacy by preventing interpretation by VLP models, and one that retains full semantic information for legitimate use—is a novel idea. The use of Conditional Latent Trigger Generation and Uncertainty-Aware Encryption-Oriented optimization adds unique elements to the field of privacy-enhancing image compression.

**Questions For Authors:**

1. What specific steps would be needed to extend the PSIC method to other data types, such as audio or video, and how do you foresee the challenges related to temporal or sequential data?

2. How does the UAEO optimization function contribute to the robustness of the PSIC method against more sophisticated attacks, such as model inversion attacks? Could further defense mechanisms be incorporated to strengthen the security guarantees?

3. Could you elaborate on the trade-offs between encryption efficiency and compression performance? In particular, how does the PSIC method compare with other state-of-the-art image compression methods, such as JPEG or WebP, in terms of file size and compression time?

4. Can you provide a deeper discussion on the potential limitations of the CLTG module, particularly in cases where the image content varies significantly from the training data? How might the system adapt to these situations?

**Relation To Broader Scientific Literature:**

This paper addresses the growing privacy concerns regarding VLP models and their ability to exploit publicly available visual data. Given the increasing reliance on machine learning models that use large-scale datasets for training, the proposed method provides a timely and important contribution to privacy-preserving data compression.

**Theoretical Claims:**

Correct.

---

> ### Author Rebuttal · Authors · 2025-04-01
>
> First of all, we deeply appreciate your kind suggestions and positive feedback on our work, which have greatly encouraged us. We hope our responses below adequately address your concerns.
>
> Q.1: Discuss the challenges of extending to other data types.
>
> Ans.: Thanks for your insightful comments. Based on our experience, we believe the proposed PSIC scheme can be readily extended to different data types, especially video. In particular, end-to-end video compression follows a frame-by-frame pipeline, with a framework similar to image compression. Hence, the CLTG module, UAEO function, and associated optimization strategy can be directly integrated.
>
> Meanwhile, certain adjustments are indeed necessary to handle domain-specific structures (e.g., temporal dependencies). Specifically, in the video compression process, the bitstream for a given frame must be reconstructed and used as a contextual prior for compressing the subsequent frame, thereby removing temporal redundancy. Taking both compression and encryption needs into consideration, we recommend decoding the bitstream into the full version, as it retains more similarity to the next frame.
>
> Q.2: Discuss the UAEO’s contribution against more sophisticated attacks.
>
> Ans.: Thanks for your suggestions. The UAEO potentially enhances the robustness of the PSIC method against sophisticated attacks, e.g., model inversion, by explicitly leveraging the uncertainty within VLP models and incorporating this uncertainty into corresponding constraints or loss functions. Specifically, UAEO employs Dempster-Shafer Theory to identify image-text pairs with high uncertainty (low confidence), guiding the embedding of targeted yet nearly invisible adversarial patterns during the image compression process. This approach potentially complicates an attacker's ability to reconstruct sensitive semantic information, thereby offering stronger privacy protection.
>
> Additionally, as UAEO primarily focuses on mechanisms related to representation-level information, further complementary defense strategies could potentially be incorporated to strengthen the overall security guarantees of PSIC. Potential enhancements include adopting differential privacy techniques (e.g., “Gaussian differential privacy”, JRSS2022) to introduce controlled data obfuscation, employing multi-modal encryption methods for broader protection against cross-modal inference threats (e.g., “BadCLIP: Trigger-Aware Prompt Learning for Backdoor Attacks on CLIP”, CVPR2024), and regularly conducting security audits combined with adversarial retraining, which may help continuously maintain or further enhance robustness over time.
>
> Q.3: Discuss the trade-offs between encryption efficiency and compression performance.
>
> Ans.: Thanks for your comments. To illustrate the trade-offs between encryption and compression performance, we present how these metrics evolve during the second training stage, where the encoder is frozen (i.e., the resulting bitstreams remain unchanged). Specifically, we obtained checkpoints (bpp=0.13) at different epochs, and evaluated their encryption efficiency and compression performance. All test settings are identical to those described in our manuscript.
> |Epoch|10|20|30|40|
> |-|-|-|-|-|
> |PSNR|24.82|24.73|24.65|24.63|
> |ASR(i2t)|0.11|0.14|0.25|0.30|
> |ASR(t2i)|0.15|0.18|0.26|0.32|
>
> As you insightfully noted, there are indeed trade-offs between these two inherently conflicting objectives: increasing encryption efficiency comes at the expense of compression performance. Nevertheless, with the help of our CLTG module, PSIC achieves a favorable balance—maintaining compression performance comparable to standard codecs while providing significantly improved encryption strength.
>
> Q.4: Compare PSIC with JPEG or WebP.
>
> Ans.: Thanks for your suggestions. The comparison results on the Kodak testing set are shown below. For a fair comparison of file size (indicated by bpp), we adjusted the quantization parameters of JPEG and WebP to achieve the same PSNR (25.85 dB) as our PSIC. Our PSIC runs on an NVIDIA 3090 GPU, while JPEG and WebP are run on an Intel Xeon Silver 4310 CPU.
> |Method|BPP|Avg Encoding / Avg Decoding (ms)|
> |-|-|-|
> |PSIC (full version)|0.13|246/30|
> |JPEG|0.28|150/2|
> |WebP|0.11|50/4|
>
> Q.5 Discuss the CLTG module’s generalization capacity.
>
> Ans.: Thanks for your insightful suggestions. First, we would like to point out that the generalization performance of the CLTG module is CLOSELY TIED to the target DOWNSTREAM mode. To ensure strong generalization, we adopt the CLIP model as the target, as it has been trained on a large-scale dataset (4 billion text-image pairs). As a result, the proposed PSIC method exhibits promising generalization ability across a variety of tasks. In our experiments, the test sets used for image classification, facial attribute analysis, and image captioning were entirely UNSEEN during training, yet our method achieved strong performance in both compression and encryption metrics.

---

### Official Review · Reviewer_ZrT6 · 2025-03-14

**Overall Recommendation:** 3

**Summary:**

This paper proposes Privacy-Shielded Image Compression (PSIC), a learned image compression framework that prevents Vision-Language Pretrained models from extracting semantic information while preserving perceptual quality. PSIC enables a single bitstream to be decoded into an encrypted version or a full version conditioned on user input. The method introduces a Conditional Latent Trigger Generation (CLTG) module and an Uncertainty-Aware Encryption-Oriented (UAEO) optimization function to enhance flexibility and attack effectiveness. Experimental results show that PSIC effectively disrupts VLP-based tasks, such as image-text retrieval and classification, while maintaining comparable compression quality to baseline models under the full version.

**Claims And Evidence:**

The effectiveness of the proposed methods requires more ablation studies, such as the Conditional Latent Trigger Generation Module and the Adaptive Multi-Objective Optimization Strategy.

**Essential References Not Discussed:**

No.

**Experimental Designs Or Analyses:**

1.	I recommend the authors to add experiments on more effective baselines like ELIC[1], TIC[2].
2.	The authors should add ablations on the proposed Conditional Latent Trigger Generation Module and Adaptive Multi-Objective Optimization Strategy. In this paper, the authors only provide ablation results on proposed UAEO optimization.

[1] He D, Yang Z, Peng W, et al. Elic: Efficient learned image compression with unevenly grouped space-channel contextual adaptive coding[C]//Proceedings of the IEEE/CVF Conference on Computer Vision and Pattern Recognition. 2022: 5718-5727.
[2] M. Lu, P. Guo, H. Shi, C. Cao and Z. Ma, "Transformer-based Image Compression," 2022 Data Compression Conference (DCC), Snowbird, UT, USA, 2022, pp. 469-469.

**Methods And Evaluation Criteria:**

Yes, I believe that introducing uncertainty to constrain image-text matching is a reasonable approach and can more effectively attack VLP (Vision-Language Pretraining) models.

**Other Comments Or Suggestions:**

Typos:
1.	Line 099, "pravicy" → privacy.
2.	Line 244, "paramterized" → parameterized
3.	Line 246, "of for the image-text pair" → of the image-text pair
4.	Line 310, unmatched number of parentheses in the equation S 1-1.

**Other Strengths And Weaknesses:**

Please refer to above comments.

**Questions For Authors:**

If the VLP model is not CLIP, is the proposed method still applicable?

**Relation To Broader Scientific Literature:**

The author considers incorporating backdoor attacks into neural image compression for VLP models, which has good practical applicability for user privacy protection on social media.

**Theoretical Claims:**

No theoretical proof.

---

> ### Author Rebuttal · Authors · 2025-04-01
>
> First of all, we would like to express our sincere gratitude for your responsible and constructive comments, which have been very helpful in further improving the quality of our manuscript. Below, we summarize your concerns into five key points and address them one by one. We hope our responses satisfactorily address all of your concerns. If there is any misunderstanding on our part, we would greatly appreciate any further clarification or additional suggestions.
>
> Q.1: Ablation study on the CLTG module.
>
> Ans.: We provide an ablation study on CLTG. We remove all CLTG modules and train only the compression backbone using UAEO-based loss function:$λ_1 L_2(x, \widehat{x}) + λ_2 L_{UC} (\widehat{x}) + r$, $\widehat{x}$ is the reconstructed image, $r$ is the bitrate. For a fair comparison, we adjusted the values of $\lambda_1$ and $\lambda_2$ so that the ablated version achieves the same ASR level as the full version. Due to the time constraint, we only provide comparison results at 0.20 BPP points shown below, and we will complete the ablation study in our revised paper.
>
> |Bpp|Meth.|PSNR|i2t|ASR(i2t)|t2i|ASR(t2i)|
> |-|-|-|-|-|-|-|
> |0.20|Ours($x_f$)|26.07|0.32|-|0.20|-|
> ||Ours($x_e$)|25.61|0.13|0.69|0.07|0.78|
> ||*w/o* CLTG|24.48|0.11|0.73|0.08|0.75|
>
> As shown, the CLTG module’s effectiveness is evident: it significantly boosts the perceptual quality (1.59dB PSNR) at the same ASR level.
>
> Q.2: Ablation study on the multi-stage training strategy.
>
> Ans.: We follow the nice suggestion to look into the effect of our multi-stage training strategy. In particular, we omit Stage 2 and instead train the model using the settings from Stage 1 until convergence (an additional 100 epochs). We present the comparison results regarding encryption performance (encrypted version) and perceptual quality (full version) below.
>
> |Bpp|Meth.|ASR(i2t)|ASR(t2i)|PSNR($x_f$/$x_e$)|
> |-|-|-|-|-|
> |0.13|Ours|0.66|0.74|24.72/24.57|
> ||*w/o* stg2|0.05|0.06|24.70/24.68|
> |0.20|Ours|0.69|0.78|26.07/25.61|
> ||*w/o* stg2|0.03|0.05|26.02/26.02|
> |0.30|Ours|0.74|0.83|27.37/25.67|
> ||*w/o* stg2|0.04|0.06|27.28/27.27|
> |0.46|Ours|0.82|0.88|28.78/27.34|
> ||*w/o* stg2|0.02|0.04|28.84/28.84|
>
> As we stated in our manuscript, Stage 2 emphasizes enhancing the capacity for divergent representation. Without this stage, encryption performance would decrease by about 72% on average in terms of ASR.
>
> Q.3 More effective compression baselines.
>
> Ans.: Thanks for your constructive comments. We have implemented the PSIC scheme in two cutting-edge LIC models (ELIC, and MLIC++). Due to time constraints, we are currently providing comparison results at a single BPP point (see below). All training configurations were kept strictly consistent with those in our manuscript to ensure fair comparisons. We will include comprehensive results on additional LIC baselines (including TIC) across multiple BPP points in the supplementary material of our revised paper. We appreciate your understanding.
>
> ||Meth.|PSNR|i2t|t2i|ASR(i2t)|ASR(t2i)|
> |-|-|-|-|-|-|-|
> |ELIC(bpp=0.19)|Baseline|27.24|0.32|0.22|-|-|
> ||Ours($x_e$)|26.64|0.30|0.19|-|-|
> ||Ours($x_f$)|26.05|0.12|0.07|0.74|0.79|
> |MLIC++(bpp=0.18)|Baseline|27.26|0.35|0.22|-|-|
> ||Ours($x_e$)|26.93|0.32|0.21|-|-|
> ||Ours($x_f$)|26.44|0.13|0.09|0.69|0.73|
>
> As shown, the proposed PSIC scheme can be readily integrated into different LIC models, demonstrating strong effectiveness in both encryption and compression performance.
>
> Q.4 Applicability to other VLP models.
>
> Ans. : Following your insightful suggestion, we have adopted another milestone VLP model, ALIGN (“Scaling up visual and vision-language representation learning with noisy text supervision”, ICML2021), as the target. Specifically, we restructured the UAEO optimization function based on ALIGN, and trained the entire framework at a single BPP point (bpp = 0.13). The corresponding results are presented below, demonstrating the strong applicability of the proposed PSIC method to other VLP models.
>
> |Meth.|PSNR|i2t|ASR(i2t)|t2i|ASR(t2i)|
> |-|-|-|-|-|-|
> |Comp. Baseline|24.83|0.26|-|0.17|-|
> |$x_f$(CLIP)|24.72|0.24|-|0.17|-|
> |$x_e$(CLIP)|24.57|0.11|0.66|0.07|0.74|
> |$x_f$(ALIGN)|24.68|0.23|-|0.17|-|
> |$x_e$(ALIGN)|24.32|0.17|0.43|0.14|0.47|
> Q.5 Proofreading for typos.
>
> Thank you for your thorough review. We will definitely incorporate your suggestions into our revised manuscript, and we will carefully proofread the manuscript to eliminate any typos or grammatical issues.

---

> > ### Comment · Reviewer_ZrT6 · 2025-04-08
> >
> > I appreciate the authors' detailed rebuttal. The revisions addressed most of my concerns, and I will raise my score.

---

### Decision · Program_Chairs · 2025-05-01

**Decision:**

Accept (poster)

**Comment:**

This paper presents Privacy-Shielded Image Compression (PSIC), a learned image compression framework designed to preserve perceptual quality while protecting semantic privacy from Vision-Language Pretrained (VLP) models. PSIC enables a single bitstream to be decoded into an encrypted version or a full version conditioned on user input.  Key components include the Conditional Latent Trigger Generation (CLTG) module and the Uncertainty-Aware Encryption-Oriented (UAEO) optimization, which together aim to mislead VLP models while maintaining compression quality.
The paper introduces a promising and practical direction for privacy-preserving image compression, especially in the context of rising concerns over VLP model misuse. While additional experiments and refinements would strengthen the work, the core idea and demonstrated potential justify acceptance.